# Mango Fruit Detachment of Trees after Applying a Blend Composed of HNO_3_ and Charcoal Activated

**DOI:** 10.3390/plants13091216

**Published:** 2024-04-28

**Authors:** David Vargas-Cano, Federico Hahn, José Luis Rodriguez de la O, Alejandro Barrientos-Priego, Víctor Prado-Hernández

**Affiliations:** 1Agricultural Engineering and Integral Use of Water, Universidad Autónoma Chapingo, Texcoco 56230, Mexico; 2Irrigation Department, Universidad Autonoma Chapingo, Texcoco 56230, Mexico; 3Department of Plant Sciences, Universidad Autonoma Chapingo, Texcoco 56230, Mexico; 4Department of Soil Sciences, Universidad Autonoma Chapingo, Texcoco 56230, Mexico

**Keywords:** harvest, mango fruits, nitric acid, activated carbon, pedicel-peduncle

## Abstract

As young workers prefer urban labors and migrate to USA and Canada, mango harvesting is becoming scarce on Mexican coasts. This seasonal labor is becoming expensive and when many orchards produce fruit simultaneously, grower losses increase. In this research, an innovative fruit detachment method was tested after applying a viscous paste to the pedicel of mango fruits hanging in the tree. Activated carbon or charcoal (AC), was mixed with different amounts of nitric acid to provide three AC composite blends named: light, medium, and dense. The nanomaterial was applied with a brush to the fruit pedicel/peduncle taking up to 4 h before the mango fruits felt to a net below the tree canopy. Mango detachment experiments indicated that the medium blend was the most efficient in releasing the fruit, taking an average of 2 h. The dense nano-material decreased latex exudation to 7% of the fruits. Fruit maturity emerged as a crucial factor for detachment time, followed by mango weight.

## 1. Introduction

Mango (*Mangifera indica* L.) is one of the most important fruits grown worldwide, where more than 150 varieties are grown [1]. Mango evergreen trees can reach heights of over 15 m as they become mature. Each mango fruit is attached from a pedicel-peduncle that, when cut, exudates an irritating liquid to the fruit that smells like turpentine and is composed of an allergenic urushiol, 5-heptadecenylreorcinol. Fruit harvesting requires excessive labor, where pickers climb tall trees to collect mangoes manually or use bamboo sticks equipped with clippers at the end, together with a bag [2] that can handle up to four fruits. It is desirable to leave 10 cm stems to avoid the spurt of milky/resinous sap during fruit collection. Harvesting mango fruit with short-end stem retain more latex with antifungal properties, making the fruit less susceptible to stem end rot [3].

Mango mature plantations (30 years old) can produce between 10 and 25 t ha^−1^ depending on the planting zone [4]. As trees require a relatively high level of irradiance for photosynthesis and fruit yield, pruning should be considered in the orchard. Pruning maintains tree size and allows better light penetration within the canopy [5], but severe pruning requires at least three years to produce fruit again. Flower induction decreases with shading reducing terminal shoots at bearing inflorescences. Fruits growing at the top of the tree show better quality than fruits inside the canopy [6] but are more difficult to harvest. Mango fruits collected from pruned trees presented a higher amount of total soluble solids (TSS), but lower total carotenoids and titratable acidity than fruits from normal unpruned trees [5]. Fruits inside the canopy keep a greener color as they are not exposed to sunlight all day and are 5% heavier than top canopy fruits [7]. High density productivity of mango trees with 1110 trees per ha at an age of 11 years was ten times higher than the traditional 100 trees per ha with a yield of 5.9 t ha^−1^ [8]. In high density trees subjected to severe pruning, the fruit weight, length, and volume decreased [6].

Activated carbon (AC) is a microcrystalline, non-graphitic form of carbon with a porous structure. AC presents small pores that boost the surface area available for chemical reactions [9]. AC obtained from coconut shells [10] has a higher density and a finer pore size distribution and is suitable for small molecule adsorption. AC can be activated physically or chemically. The chemical reaction of AC depends on the chemical activator, and its concentration, temperature, and activation time are the parameters that determine the reaction range [11]. In order to obtain the AC from a coconut shell material it needs to be impregnated by an acid and requires high temperatures (400–600 °C) to be carbonized. Its application in agriculture includes sugarcane bagasse heavy metal adsorption [12], malathion removal [13], and soil remediation [14,15].

Seasonal workers harvest mango trees, but are becoming scarce throughout the coasts of Mexico. This situation leads to mango fruit buildup on thousands of trees, food losses, and pest development. Harvest and marketing costs have been evaluated to make up 21% of total mango production costs [16]. Machine tree harvesting methods can reduce the cost of harvesting by 35–45% [17] in flat fields. Therefore, a project was started to automatize mango collection in flat and hilly fields, with the use of drones. It is important to encounter a simple methodology that can provide an automatic detachment without removing the entire pedicel and causing fruit discoloration through sap-peel contact [18]. Detached mango fruits fall to a net just below the tree canopy and determine its time (Figure 1). The manuscript layout considers the experimental setup including the AC blend preparation in the next section. In Section 3, the fruit detachment efficiency results under different maturity stages, firmness and size are analyzed, to be further discussed in Section 4. Lastly in Section 5, future directions required to bring this innovation to practice are discussed.

## 2. Materials and Methods

The experiment was conducted in May 2023 in a commercial mango orchard located at Loma Bonita, Guerrero, Mexico (17°25′47″ N, −101°11′19″ W, 17 MASL). The mature orchard has ‘Kent’, ‘Haden’, and ‘Keitt’ mango trees. Preliminary tests were carried out with ‘Haden’ fruits (Figure 2a). The final tests were carried out with ‘Keitt’ mangoes, at different maturity stages. The botanical name for the stem attachment of the mango fruit is called pedicel and continues with the peduncle (Figure 2b).

In two different experiments, acid was applied to the fruit pedicel to evaluate whether the fruit fell after some time. In the first experiment, three different acids were applied directly to “Haden” fruit pedicels. In the second experiment, a blend with nitric acid was applied to fruits of the “Keitt” variety. In both experiments, 90 fruits were randomly selected from 10 different trees at heights ranging between 1 and 1.5 m from the floor, so 9 pedicels from different fruits per tree were treated. AC and nitric acid were mixed to provide a material that could be pasted to the fruit petiole-peduncle. This nano-structured material will be named AC (activated carbon) blend or AC paste throughout this paper. A brush applied the prepared solution to each fruit pedicel.

The mango harvesting system used a net fixed to the tree trunk. Net installation covered the area beneath the tree canopy and was secured with wood-based supports. These meshes prevent the mango fruits from hitting the ground, reducing damage from impacts.

### 2.1. Nitric Acid (HNO_3_), Phosphoric Acid (H_3_PO_4_) and Hydrochloric Acid (HCL) Application at the Pedicel-Peduncle

The idea of applying acid to the pedicel-peduncle area was to detach the fruits in the minimum possible time. In the first experiment, three different acids were applied to the fruit pedicel-peduncle to induce detachment. The acids used were 55% nitric acid (HNO_3_), 85% phosphoric acid (H_3_PO_4_), and 28% hydrochloric acid (HCL). A dropper for punctual application applied phosphoric acid to 10 mango hanging fruits (Figure 3a). Afterwards, nitric acid was applied to the pedicel of 45 Haden mango hanging fruits with a hair nano atomizer; drops were formed and rolled down, ending over the fruit surface (Figure 3b). The effectiveness and damage caused to surrounding leaves, fruit peduncle, pedicel, and fruit peel were evaluated. The main damages were caused due to the application method (Figure 3). The phosphoric and hydrochloric acid were applied, and the drops and acid runoff are shown in Figure 3c. The acid drops burnt-out the mango peel and never detached the fruit. The damage to the fruit and leaves is shown in Figure 3d,e.

Activated carbon powder was used to obtain a more consistent material that could avoid acid rollout after application. Blend preparation took place 15 min before application and consisted of 4 g of coconut activated carbon powder mixed with 4, 5, or 6 mL of 55% nitric acid (Figure 4a). These variable-consistency AC blends were named light, medium and dense, respectively. AC powder and acid were weighted with a scale (ADIR, 10501, Mexico). The activated carbon was deposited in a laboratory sample jar, and then 4 to 6 mL of 55% nitric acid was added depending on the final blend required (Figure 4b) [19]. It was mixed with a plastic stirrer until a homogeneous blend was obtained and then the container was sealed until its application time. The AC blend (acid-composite) could last for one day before its application, and during its preparation an exothermic reaction took place. It was mixed with a plastic stirrer until a homogeneous mixture was obtained and then the container was sealed. The container was shaken for 5 s using the applicator device before application. Three groups consisting of 30 “Keitt” mango fruits each were randomly selected, and the fruits were numbered from 1 to 90. Each blend was applied evenly to 30 fruits at the pedicel-peduncle with a 15-cm-long painting brush. The acid-composite covered a cross section of 1 cm, 1 cm above the pedicel abscission zone (Figure 4b). The application began at 9 am, ending 20 min later for each group.

In an additional experiment, the AC blend temperature during preparation was monitored with digital sensors (Maxim Integrated, DS18B20, San Jose, CA, USA). Three sensors recorded the temperature (Figure 5a) every 10 s for each AC blend type using a homemade datalogger based in the ESP32 microcontroller (Espressif Systems, DevKit V1, Shanghai, China). The test was performed under a constant ambient temperature of 25 °C (Figure 5). AC blends applied over the sensors are shown in Figure 5b.

### 2.2. Mango Fruit and Pedicel-Peduncle Measurements

Each pedicel-peduncle diameter from a selected hanging mango fruit was measured with a digital micrometer (Mitutoyo, QuantuMike, Naucalpan de Juárez, Mexico) before the AC paste was applied. After applying the blend to the fruit hanging from the tree, the time until the fruit fell was obtained. An infrared camera (FlirC5, Teledyne FLIR, Wilsonville, OR, USA) was used to measure the temperature at the pedicel once the AC blend was applied. The digital camera presents 5 MP meanwhile the IR presents 160 × 120 infrared pixels and can measure temperatures between 0 and 100 °C. Once the mango fruits were collected over the net, several measurements were obtained (Figure 6). Weight was measured with a digital scale (model BAREC-5A, Rhino, México City), while the percentage of fruits affected by latex and/or acid was evaluated by visual inspection. Firmness was measured using an analogue penetrometer (model GY-03, LUZEREN^®^, México) equipped with an 8 mm insertion tip. Total soluble solids given in °Brix were quantified with a portable refractometer (model 300060, Sper Scientific Ltd., Scottsdale, AZ, USA). These measurements were taken once the fruit fell from the tree.

Pedicel-peduncle samples exposed to the reactive mixture were collected, as well as those randomly selected control samples. The samples were immersed in a solution composed of glycerin and alcohol in a ratio of two to one, to ensure their preservation and transportation to the laboratory. Afterwards, in the laboratory longitudinal histological sections were obtained with a sharp bistoury to observe and compare the effects of the acid application. Pedicel-peduncle samples of approximately 2 cm were fixed in 96% ethanol: 100% glacial acetic acid (2:1; *v*:*v*) and processed in an automatic tissue exchanger (Tissuematon Fisher) with 2-ethoxyethanol (cellosolve) and xylene. Finally, the tissue samples were transferred to paraffin (55 °C) staying 72 h inside a stove. The paraffin pyramid was prepared according to Sass (1968) technique [20] in a rotary microtome (model 820, American Optical, Vernon Hills, IL, USA), before obtaining 10 µm thick longitudinal slices. The sliced sections were stained for 30 min at room temperature in a mixture of equal volumes of 0.1% aqueous solutions of safranin and fast green, then washed in distilled water for 5 min and washed twice with absolute alcohol for 2–3 min [21]. The stained sections were then mounted on slides with coverslips using Haupt adhesive and 10% formalin [20]. Digital images were obtained and saved in JPEG format with the aid of a Motic B3 Professional Series microscope, with the adaptation of a Moticam 480 camera with a 16 mm adapter.

### 2.3. Data Analysis

The data were analyzed using the RStudio software, version 4.2. Fruit weight, stem diameter, soluble solids content (SSC), and firmness were subjected to the Kruskal–Wallis test to determine statistically significant differences between treatments [22,23,24]. A survival analysis was used to explore fruit cracking difference time. It used the “survival” package and the non-parametric Kaplan–Meier (product limit) estimator defined by Equation (1) [25,26,27].
(1)$$\hat{S}\left(t\right)=\prod_{i:t_{i}\leq t}\left(1-\frac{d_{i}}{n_{i}}\right)$$
where: $\hat{S}\left(t\right)$ is the Kaplan–Meier estimator, $t_{i}$ is the time at which the event occurs, $d_{i}$ is the number of events that occurred at time $t_{i}$, and $n_{i}$ the number of events after time $t_{i}$.

Survival function comparison was carried out using Log-Rank tests. It evaluated significant differences in the fruit detachment time after different blends were applied. Additional hypothesis tests were carried out to evaluate the specific influence of various parameters, such as soluble solids content (SSC), mango fruit weight, stem diameter, and fruit firmness, on fruit detachment success under different acid–composite materials applied to the fruit pedicel.

All statistical tests were performed with a significance level α = 0.05, assuring results robustness. In cases where *p*-value adjustments were made using the “Bonferroni” method, a corrected significance level was used $\left(\alpha =0.05/n_{comparaciones}\right)$. This practice allows rigorous control over type I errors [28], strengthening the validity and reliability of the results.

## 3. Results

Nitric acid was the only acid that detached the fruit during the first experiment and it occurred after a period of 4 days. As severe damage was caused to peel of all the fruits, a substance capable of fixing the acid to the peduncle had to be added. Fruit damage was reduced, meanwhile acid performance improved due to the punctual contact. Glycerin was used as an acid-binding agent to avoid acid drainage, inducing fruit detachment after 24 h, but peel damage was still observed. The activated carbon AC blend produced less damage and fruit separation occurred in a lower time of 7 h.

### 3.1. AC Blend

Toxic vapors are released during AC-blend preparation caused by an exothermic reaction. After mixing the nitric acid with the activated carbon, the AC paste temperature rises rapidly (Figure 7a). The light paste reaches the highest temperature of 29.7 °C after one second (Figure 7b) with the ambient temperature being 25 °C. Subsequently, it drops after the first hour and stabilizes after 2 h (Figure 7b). After application, the temperature peak occurs for the light, medium, and dense blends after 10, 20, and 40 s, respectively.

The nano-paste inside the bottle remains at a stable temperature, increasing when it is shaken, even up to a day after being prepared. Fruit surface temperature depends on fruit transpiration and vapor pressure deficit (VPD) between the its surface and the air. High temperature reduces fruit acidity and sugar content with dry matter accumulation [29]. A sunny exposed fruit in the outer-canopy will present a lower VPD than shaded fruits [30]. The ticker cuticle layer in mangoes grown within the canopy surrounding, changed its cuticle conductance. This explains why the mango tissue infrared temperature was higher than 34.5 °C. The pedicel (Figure 8a) measured 32 °C, four degrees over the AC paste temperature as solar irradiation heats the pedicel. The AC-blend applied to the pedicel remains at its temperature, being lower than the fruit temperature. Mango abscission and pedicel temperatures are shown in Figure 8b and 8c, respectively. In Figure 8c the temperature of the pedicel-peduncle is of 31 °C once that is detached from the fruit.

### 3.2. Pedicel

The fruits were detached between the pedicel and peduncle, above the mango abscission zone, as observed in Figure 9a,f. The large periclinally-elongated cells of the peduncle suffer deterioration when coming into direct contact with the nanocomposite blend, resulting in notable damage to the internal tissues (Figure 9b,c). The damage ends with the supporting capacity of the fruit, causing the pith rupture (Figure 9e) and the fruit detachment. The main indicator of peduncle cell degradation is revealed when latex flows out from the lactiferous channels. This latex can fall onto the fruit, taking the nitric acid over the peel (Figure 9f). These events cause substantial fruit damage (Figure 9d).

In the longitudinal histological cuts of the control pedicel-peduncle and those treated with the AC blend (Figure 10), it is evident that damage is caused by the nanocomposite solution, extending throughout the entire application region. The external peduncle tissue exhibit cut that reach the internal tissues in areas other than the main cut and is attributed to the action of the mixture AC blend even after the fruit has been detached. The tissue damage goes into the xylem layer, that includes the layers having the laticifers that conduct the latex. In each of the four images, the left section is the one fixed to the fruit and the letter. Only a few letters were placed in Figure 10, to show the different tissues and avoid data agglomeration. In the upper-right image, the microphotography shows the effect after applying a light AC paste, meanwhile in the lower-right a dense AC paste damages more the pedicel.

### 3.3. Firmness, TSS and Weight Measurements

The results obtained with the Kruskal–Wallis tests [22,23,24] revealed the absence of significant differences between groups (Table 1). The presence of outliers in mango weight did not affect the analysis given its robustness. Moomin (2021) reported an average of 6.56°Brix in ‘Keitt’ mangoes at its physiological maturity or early harvest stage [31], so our average total soluble solids, TSS is similar in each of the three fruit groups. Firmness of many fruits exceeded the threshold of 11 kgcm^−2^, agreeing with the findings discussed by Siller-Cepeda et al., 2009 [32].

### 3.4. Fruit Detachment

Fruit detachment was uniform, regardless of the AC paste used. However, significant variations were observed in the time elapsed from blend application to fall. Medium AC blend detached the fruit in an average time of 2 h, meanwhile the light blend detachment time had an average of 3 h.

The dense AC blend required more time than the other two pastes to detach the fruits from the branches. The fruits started falling after one hour and some of them required up to 6 h. Fruit damage caused by the latex fluid with the dense AC blend was lower than with the other AC composites. Higher latex flow was adsorbed at sunrise when there is greater sap flow due to high water content or turgor pressure [33]. The light paste caused maximum damage to the fruits (Figure 11).

Figure 12a shows the Kaplan–Meier curves corresponding to the each of the three pastes being applied [25,26,27]. The stepped lines in the graph represent the probability of survival, interpreted as the possibility that some fruit remain hanging at any moment after paste application. This graphical approach, based on the non-parametric Kaplan–Meier estimator, offers a precise and dynamic visualization of the detachment rate over time, allowing variations in the effectiveness of treatments to be discerned and highlighting critical moments in the fruit falling process. A probability of zero indicates that all fruits fall, meanwhile a curve that descends quickly until a value of zero indicates that the AC blend requires a shorter time to induce fruit detachment. This graphic behavior highlights the effectiveness and speed between pastes.

In addition to the Kaplan–Meier survival analysis, another valuable outlook consists in interpreting the hazard function [25,26,27]. This function not only provides an accurate representation of the instantaneous risk rate but it also explains the dynamics of the process. It also provides an adequate approximation to the event incidence rate [34]. The hazard function curves of the different AC pastes showed a steep positive slope (Figure 12b), which indicated a rapid increasing hazard rate. This pattern suggests that the medium AC blend is more effective than the others pastes. Risk functions also highlight critical points, where changes are more pronounced. A peak on the graph indicates a high fruit detachment probability, meanwhile a valley suggests a low probability. In the second hour after blend application, a critical point for the three pastes was observed. In this period, the greatest number of fruit detachment occurred. This detailed analysis with risk functions contributes to identify key moments in the separation of the fruit from the stalk. Additionally, it helps to analyze each AC blend efficiency to obtain a comparative evaluation.

### 3.5. Acid-Activated Carbon Performance

The Log-Rank test was used to compare the treatment’s performance. The results revealed that there are statistically significant differences between the three AC blends used (*p*-value = 0.00001), indicating that at least a couple of them present significant differences in terms of fruit separation, Table 2. The application of the Log-Rank test with *p*-value adjustment, by using the Bonferroni method allowed us to identify statistically significant differences between the light and dense treatments (*p*-value = 0.0036). Differences were also found between the medium and dense treatments (*p*-value = 0.000035). This was evidenced when comparing the *p*-values obtained through the test with the corrected significance value $\left(\alpha =0.05/3=0.0167\right)$. However, no significant differences were observed between the light and medium AC blends (*p*-value = 0.1648). The integration of these results together with the graphical representation of survival supports that the medium AC blend presents the lowest survival rate. Therefore, the medium blend detaches fruit faster.

#### 3.5.1. Effect of Ripeness in AC Blend Fruit Detachment

Fruits classified as ripe or unripe based on a TSS sorting criterion requires of minimum acceptable value of 6.6°Brix. This value is required to accept mango fruits by packinghouses according to the Mexican standard [35] (NMX-FF-058-SCFI-2006) and nearby 6.56°Brix mentioned by Moomin et al. (2021) [31]. With this classification threshold, we found that 47.8% of the total fruits were immature.

The general test, without considering a particular AC blend, revealed a *p*-value of 0.03. After adjusting the *p*-value with the Bonferroni method, a new *p*-value of 0.031 was obtained, that exceeds the corrected significance value (α = 0.05⁄2 = 0.025). This finding, together with the significant difference between treatments, suggests exploring together the blend type and the fruit maturity variable. With the Log-Rank test, a *p*-value = 0.003 was obtained, indicating that in at least one blend, the maturity influenced detachment timing, so separate tests were performed. The results showed that fruit maturity stage does not have a significant effect on fruit removal when light and medium paste were used. However, a statistically significant difference was observed with the dense AC blend (*p*-value = 0.032). The slope of immature fruits decreases rapidly, indicating that detachment occurs in considerably less time (Figure 13). This pattern suggests that the dense AC blend presents a more pronounced impact on immature fruit detachment than with light and medium paste material.

#### 3.5.2. Effect of Firmness, Diameter, and Weight in AC Blend Fruit Detachment

The results of all the tests are summarized in Table 2. As mentioned earlier, maturity stage is the variable that has the greatest influence in fruit detachment under the different AC blend treatments. The second variable with greater impact is weight (Figure 14). Fruit weight was classified into three categories before its analysis (Table 2). Weight showed a specific influence when the dense AC blend was applied.

The peduncle diameter variable showed a significantly low *p*-value, but still above the significance threshold. Pedicel diameter was positively correlated to fruit size and can limit fruit transpiration and water movement if it is small [36]. However, accepting the hypothesis that there are no significant differences could result in a type II error. In an ideal scenario, where the blend is precisely applied in each area, a larger diameter would require of more paste material. In this case the treatment would be more effective. Fruit firmness had no statistically significant differences for fruit detachment under the three blends.

## 4. Discussion

Physical-mechanical properties of the fruit-pedicel-peduncle system showed that the resonance frequencies of green and ripe fruits could be used for selective harvesting [37]. Fruits being harvested fall to the floor and affects its quality, as bruises result from the impact [38]. In experiments with green mature “Carabao” mango fruits in the Philippines, 80% of the fruit that fell from a height of 5 m suffered cracks and 20% were bruised [39]. Fruit impacts result in cell wall and membrane rupture [40]. In our experiment, the mesh avoided any cracks or bruises. Although companies develop automated and robotic harvester systems, they are costly for small farmers and difficult to employ in places within hills and mountains. Manual and robotic fruit manipulators have end-effectors or grippers to handle the fruits carefully [41] but operate slowly. During harvest, almost half of the fruits to be collected are at the right maturity level while the rest are either over-ripened or under-ripe [42]. Mangoes collected at their right ripeness will have the best nutritional content and quality, increasing its marketing price.

Mango have ducts called laticifers, which transport sap (latex). In the fruit abscission zone, these ducts are found within the fruit duct and at the pedicel-peduncle, but do not interconnect between them [33]. A smaller amount of latex ducts was found in the pedicel but larger in size [43,44]. Near the peel, canals were thinner than those closer to the pulp. The acid sap (pH = 4.3) under considerable pressure, is released from the ducts causing mango peel damage [45]. Peel sapburn injury reduces external fruit quality and can be controlled by applying sodium hydroxide [46]. Pedicel retention after paste application and injuring the tissue retained the latex in the fruit and avoid sap damage. From the 90 fruits pasted with AC blend, only 23 showed sapburn symptoms. The acid-composite applied to each fruit was not always the same and could affect detachment results. Other possible benefit from the “chemical cauterization” could be anthracnose incidence that has been found to decrease when latex is retained at harvest [43], so with AC blend application may have another mango postharvest benefit.

## 5. Conclusions

Seasonal labor is expensive and difficult to obtain when there are several orchards within a nearby area producing mangoes at the same time. An innovative fruit detachment method used an emulsion produced by activated carbon and nitric acid which was applied to the peduncle-pedicel of mango fruits hanging in the tree. The application of this blend was more effective than when nitric, phosphoric, and hydrochloric acids were applied alone. Activated carbon (AC), was mixed with different amounts of nitric acid to provide three AC composite blends of different viscosity, and named light, medium, and dense. The nanomaterial was applied with a brush to the fruit peduncle-pedicel taking up to 4 h before the mango fruits felt to a net below the tree canopy. The medium blend was the most efficient in releasing the fruit, taking a minimum time of 2 h. Fruit maturity status resulted in the most predominant factor on fruit separation when treatments with different blend consistencies were used. The light emulsion provided the slowest detachment performance causing staining of the fruit with latex and diminishing mango quality. This medium blend could be an alternative for harvesting mango fruit in the future in an automatically with the help of UAVs.

## 6. Future Work

In future work it is necessary to apply the technique to an entire farm. Pruning technologies must be evaluated as well as vision AI (artificial intelligence) tools in order to apply the AC blend automatically. The latter will include the use of various techniques, including robots and drones.

### 6.1. Canopy and Fruit Management

Mango trees have many different architectures according to their size and density. Mango tree management requires of a better knowledge of its architecture and its impact on the growing units (GUs) [47,48]. Plant architecture development given by GUs and inflorescences are dependent on the environmental variables of temperature, relative humidity, and solar radiation. In dense mango canopies, shading results in poor floral initiation [48]. Maybe 20% of fruit production is within panicles located in the interior of the tree, making paste application difficult for aerial vehicles. Therefore, special pruning and high-density orchards are required for automatic detaching of mango fruits. Trees maintain a balance between leaves and fruits when pruned letting an ideal sunlight penetration [49], where a flat canopy can be optimal for drone use, to apply the AC blend. Fruit drop can be reduced by applying the right amount of fertilizer, resulting in more fruits per panicle [50]. Pollination and distance between trees present an important variable for fruit production per panicle [51]. The quantity of terminal canopy per square meter shows differences according on plant canopy position and plant side interaction [51]. In both east and west tree sides, higher number of panicles were found in the bottom and medium section of the canopy [52].

### 6.2. Technology for Fruit Detection

The AC blend must be applied precisely to each mango hanging in the tree. Machine vision technology (Figure 15) has been used every day more to quantify fruit, vegetable, and crop yields through unmanned aerial vehicles (UAVs), satellites, or ground equipment [53]. This technology can detect the fruit and its pedicel accurately with RGB cameras [53,54,55]. Imaging conditions within an orchard are poorly controlled, so its quality is affected by the illumination conditions [56]. CNN segmentation [57] networks and 3D-segmentation [58] are capable of separating sections of any object class. Color can be used as a criterion in 3D segmentation, but it is much slower than 2D segmentation. Whichever platform is used, soil borne or UAV, images should be acquired and processed before applying the AC blend to the fruit pedicel-peduncles. In Figure 15, the block diagram shows image acquisition from ground or aerial platforms, so that after segmentation and peduncle detection, the AC blend can be positioned precisely by contact or remotely.

## 7. Patents

A patent (MX/a/2024/004431) is being introduced to IMPI which is the Mexican intellectual right office with name: “Método para inducir el desprendimiento de frutos mediante aplicación de una emulsión ácida” [Method and composition to induce the detachment of mango fruits from trees and other similar fruits]. Reference [19] acknowledges this patent.

## Figures and Tables

**Figure 1 plants-13-01216-f001:**
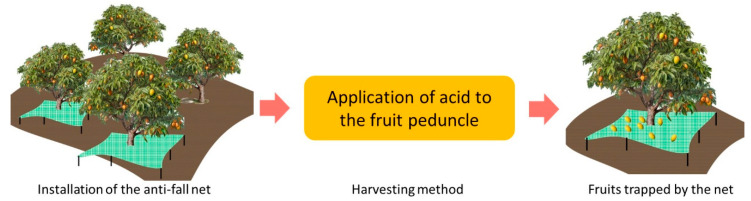
Net structure catches fruits falling from the tree.

**Figure 2 plants-13-01216-f002:**
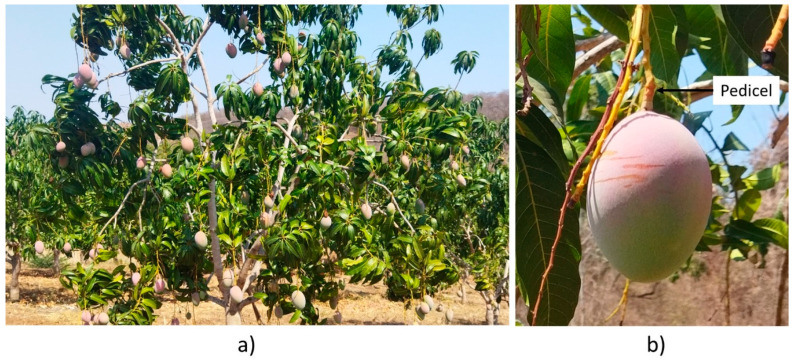
Keitt mango (**a**) fruits hanging from the tree, (**b**) showing pedicel.

**Figure 3 plants-13-01216-f003:**
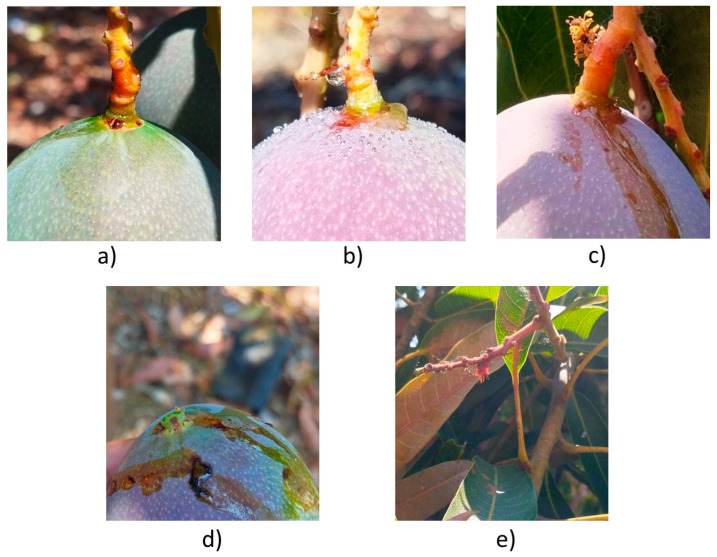
Fruit (**a**) with phosphoric acid applied at the fruit-pedicel, (**b**) drops of nitric acid after being atomized at the pedicel, (**c**) runoff of hydrochloric acid on the surface of the fruit, (**d**) fruit with severe damage caused by the atomization and runoff of nitric acid, (**e**) leaves damaged by the atomization of nitric acid.

**Figure 4 plants-13-01216-f004:**
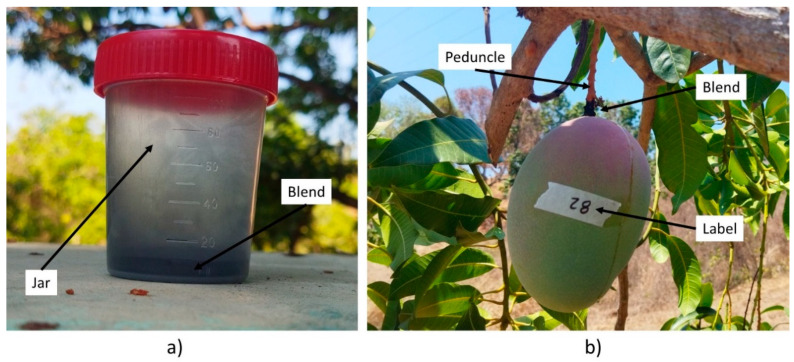
AC blend (**a**) preparation, and (**b**) application to the fruit pedicel.

**Figure 5 plants-13-01216-f005:**
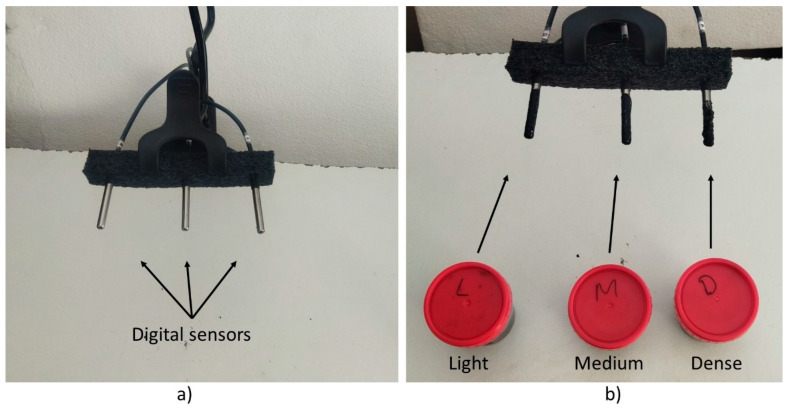
Temperature measurement (**a**) before and (**b**) after applying the three different AC blends to the sensors.

**Figure 6 plants-13-01216-f006:**
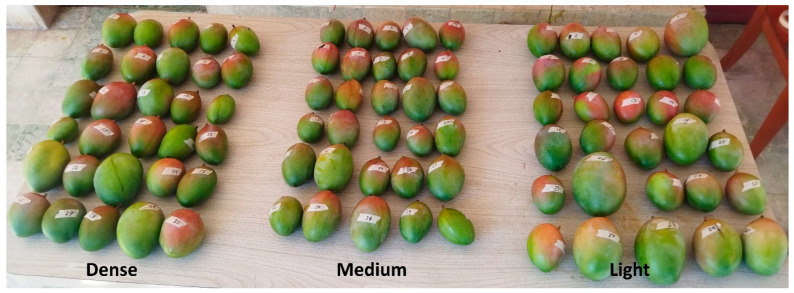
‘Keitt’ mango fruits collected at the net after pedicel breakdown.

**Figure 7 plants-13-01216-f007:**
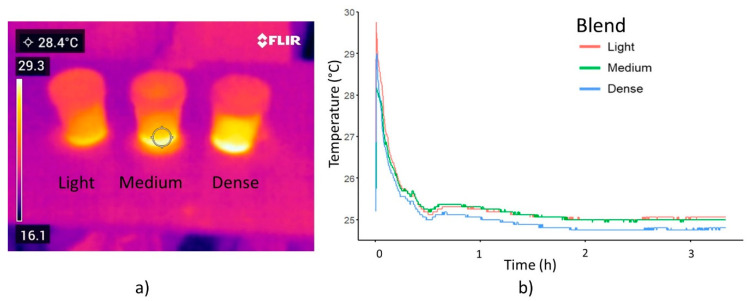
AC blend temperature (**a**) after preparation, and (**b**) during and after application to the digital sensors.

**Figure 8 plants-13-01216-f008:**
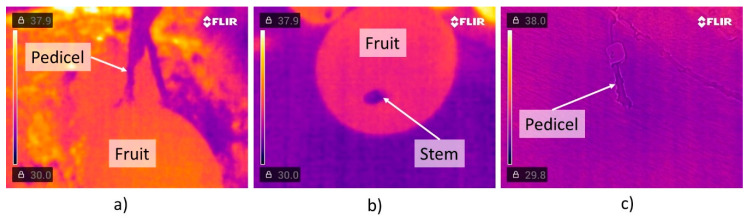
Thermal image (**a**) after applying AC blend to the fruit pedicel, (**b**) of the detached fruit, and (**c**) pedicel after fruit detachment.

**Figure 9 plants-13-01216-f009:**
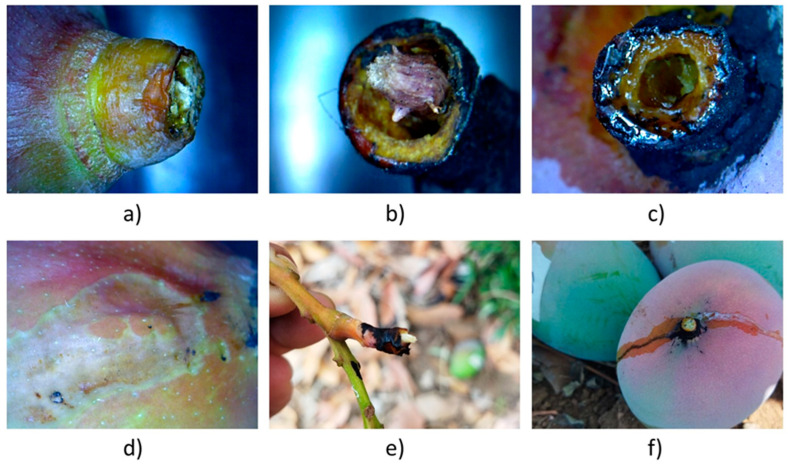
Pedicel rupture with AC blend (**a**) made at the abscission zone, (**b**) letting conducting tissues, (**c**) with bark cauterization after xylem and phloem destruction, (**d**) fruit surface burn-out due to acid runoff, (**e**) stalk end after fruit detachment, and (**f**) fruit showing acid and latex flow.

**Figure 10 plants-13-01216-f010:**
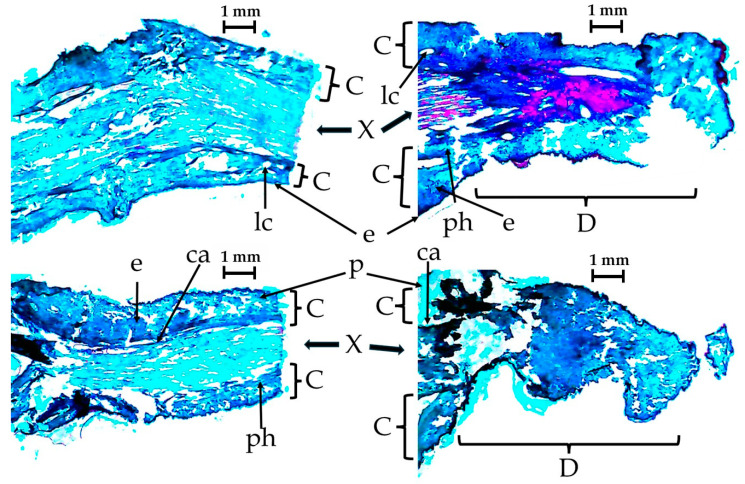
Microphotography of untreated pedicel-peduncle to the left and of treated pedicel-peduncle on the right. C: Cortex from outside to inside; epidermis (e), parenchyma (p), lactiferous channels (lc), phloem (ph), and cambium (ca). X: xylem and pith. D: damage area that includes mainly the cortex and part of the xylem, where a large constriction is evident.

**Figure 11 plants-13-01216-f011:**
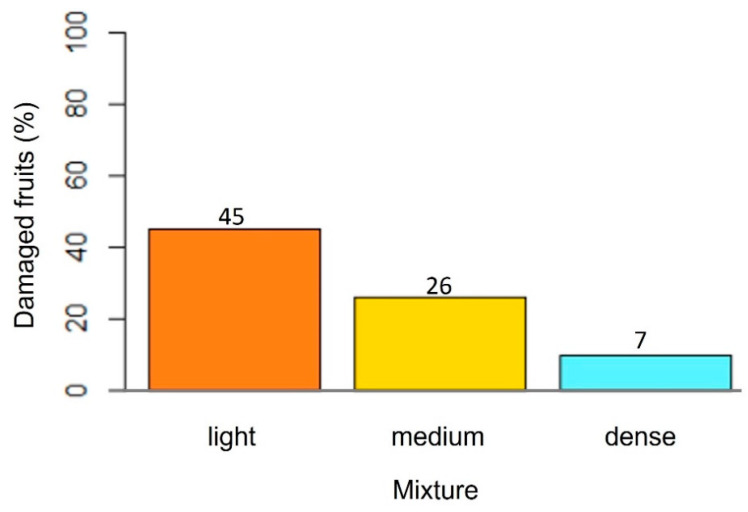
Mango fruits affected by latex or paste contact with latex.

**Figure 12 plants-13-01216-f012:**
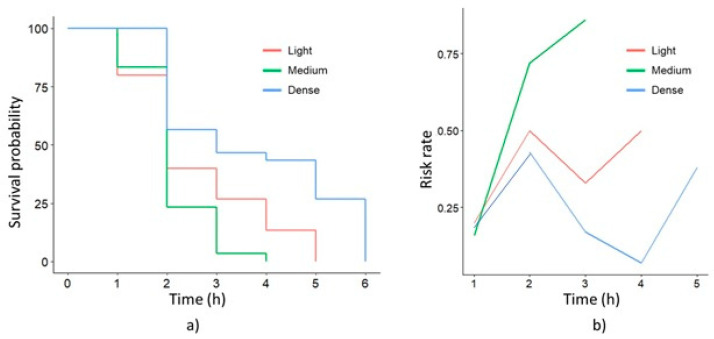
Survival analysis of the three blend treatments (**a**) with Kaplan–Meier curves, and (**b**) hazard functions.

**Figure 13 plants-13-01216-f013:**
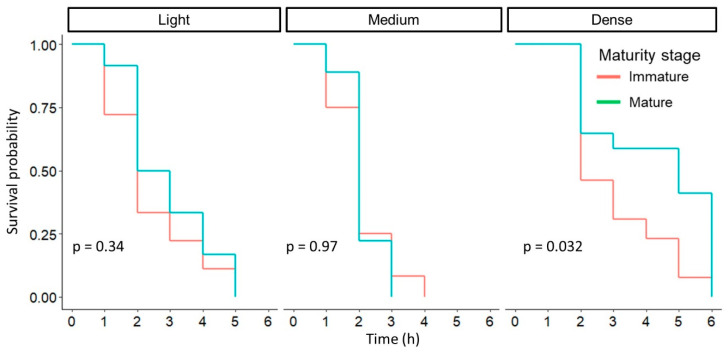
Survival plots of maturity stage and AC blends applied together with *p*-values obtained with the Log-Rank test.

**Figure 14 plants-13-01216-f014:**
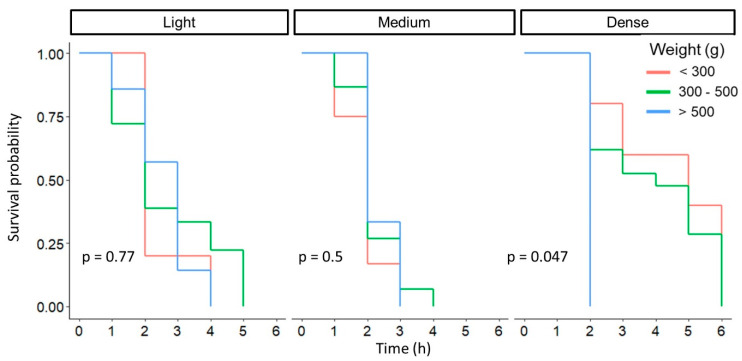
Survival plots for weight and AC blends applied together with *p*-values obtained with the Log-Rank test.

**Figure 15 plants-13-01216-f015:**
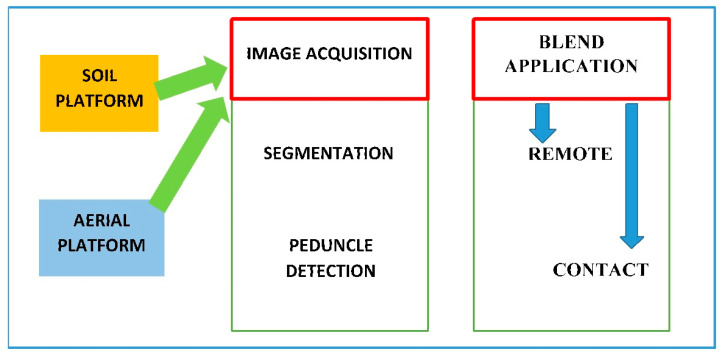
Block diagram showing technologies that will be applied to automatize the process.

**Table 1 plants-13-01216-t001:** Statistical analysis of fruit weight, diameter, TSS, and firmness, evaluated by the Kruskal–Wallis test (*p* < 0.05).

AC Blend	Statistics	Weight (g)	Diameter (mm)	TSS (°Brix)	Firmness (kgcm^−2^)
Light	Min	254.0	4.21	5.0	10.8
	Mean	442.8	4.81	6.5	17.6
	Max	1129.0	5.85	10.0	23.2
	SD	209.77	0.34	0.96	3.55
Medium	Min	211.0	3.71	4.0	14.0
	Mean	344.8	4.79	6.6	17.9
	Max	569.0	5.80	9.0	23.2
	SD	100.11	0.52	1.59	3.0
Dense	Min	182.0	4.16	5.0	14.8
	Mean	388.5	4.73	6.5	19.2
	Max	699.0	5.97	8.0	23.2
	SD	118.24	0.40	0.82	2.02
Significance	*p*-value	0.078	0.414	0.471	0.204

**Table 2 plants-13-01216-t002:** Log-Rank test results with and without Bonferroni adjustment for categorical and continuous variables.

Variable	Log-Rank *	Log-Rank + Bonferroni ^$^	Log-Rank with AC Blend *		
			Light	Medium	Dense
Maturity	0.03	Mature vs. Immature ^$$^ → 0.031	0.34	0.97	0.032
Weight (g)	0.2	<300 vs. 300–500 → 1.0 <300 vs. >500 → 1.0 300–500 vs. >500 → 0.3	0.77	0.5	0.047
Pedicel diameter (mm)	0.3	<4 vs. 4–5 → 1.0 <4 vs. >5 → 1.0 4–5 vs. >5 → 0.43	0.91	0.92	0.053
Firmness (kgcm^−2^)	0.2	<18 vs. 18–20 → 0.32 <18 vs. >20 → 0.82 18–20 vs. >20 → 1.0	0.35	0.56	0.45

Significance value * α=0.05, ^$^ Corrected value ^$^ α=0.05/3=0.0167 and ^$$^ α=0.05/2=0.025

## Data Availability

The data will be made available on request.

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
