# Peer review of "Mango Fruit Detachment of Trees after Applying a Blend Composed of HNO3 and Charcoal Activated"

_plants, 2024, doi:10.3390/plants13091216_

Round 1
Reviewer 1 Report
Comments and Suggestions for Authors
I reviewed the manuscript entitled 'Mango fruit detachment of trees after applying a blend composed of HNO3 and charcoal activated. The work turns out to be interesting and, in my opinion, it is well written. I would only have a few questions and/or curiosities:
1) In line 63 you estimate the costs (21% of total mango production costs) of harvesting and marketing and in line 64-65 you write that mechanical harvesting can reduce harvesting costs by 35-45%. Were you able to estimate the costs of manually applying the blend to the mango pedicel?
2) In line 85 (first experiment) could you please specify how many fruits and trees?
3) Same thing in line 87 (second experiment). How many trees?
4) The first citation (line 467) is missing.
Author Response
In line 63 it was estimated in Florida the harvesting costs and how mechanical harvesting can reduce costs. It was added the red words [17] in flat fields. In the next line it was added mango collection in flat and hilly fields.
In line 85 and line 87 it was added: In the first experiment, three different acids were applied to “Haden” fruit pedicels directly. In the second experiment a blend with nitric acid was applied to fruits of the "Keitt" variety. In both experiments, 90 fruits were randomly selected from 10 different trees at heights ranging between 1 and 1.5 m. Aprox 9 fruits were selected from each of 10 trees in each experiment.
The first citation in line 467 was added.
Reviewer 2 Report
Comments and Suggestions for Authors
The topic of the study is novel, and I believe it will be useful research content if applied in the field in the future.
However, there is a question as to whether the method of laying a net under the tree and detaching the fruit to drop it when harvesting mangoes can be applied in the field.
I understand that this method is mainly applied to crops with hard fruits, but please add citations or data that show that this method is or will be applied to mango.
Line 73: Does Fig 1 suggest this method?
In the case of mango, the method in fig 1 may be considered when the tree is old and tall and it is difficult to reach the fruit when harvesting by hand.
However, in this case, disposing the acid into the pedicel would seem to be equally difficult(Line 97). What do you think?
Line 212: In Figure 8, C, the temperature index is displayed in yellow up to 38 degrees, but in Figure C, the yellow color does not appear. Please edit (enlarge?) the photo so that the part that appears up to 38 degrees appears in the picture.
Line 238: There is no information about microphotography in “Materials and Methods” or in the description of Fig. 10. I also need an explanation of what the top and bottom two photos are.
Line 315: Please insert data source,, it may be table 2?
Line 366~413: Most of them look like an introduction, not a discussion about the results.
Line 367-381: Discussions are often about what is generally known rather than about actual results.
Line 370~371: This discussion sentence was not found in reference 33.
Line 414~428: The 4.2 section did not look like it related to these experiments.:
Line 419: What is UAV?

Minor editing of English language required
Author Response
The paper was revised and the following points addressed:
PLEASE CHECK pdf FOR COLOR TEXT
In the first question the net is used to avoid that the fruits hit the hard soil and their quality gets damaged. I t has been applies for nuts in the field in harvesting machines that work by vibration. The nuts fall to a net, put together and finally moved to a big container. Until now there are no applications were these this nets are used for stone fruit picking. I saw some robots that pick oranges and turn the fruit and once in the hand it lets it fall to a container but each fruit hits the next.
In our experiment we wanted to take care that the fruits didn’t suffer any damage. Yes, it is useful for tall trees as high density trees only grow up to 3 m.
We brought some comments that we have in the first paragraph of the discussion and added in a new paragraph in line 93. The mango harvesting system used a net fixed to the tree trunk. Net installation covered the area beneath the tree canopy and was secured with wood-based supports. These meshes prevent that mango fruits hit the ground, reducing damage by impacts.
Concerning to your question about disposing the acid into the pedicel would seem difficult (line 97)…Manually was simple because we reach the fruits easily. We are working with the UAV program so that the pencil can take the AC blend and reach the pedicel. We didn’t mention this because we are still working in it.
In Figure 8C the temperature index is displayed in yellow. Temperature is lower and 35°C does not appear.
Fruit surface temperature depends on fruit transpiration and vapor pressure deficit (VPD) between the its surface and the air. High temperature reduces fruit acidity and high sugar content with dry matter accumulation [29]. A sunny exposed fruit in the outer-canopy will present a lower VPD than shaded fruits [30]. The ticker cuticle layer in mangoes grown at the canopy surrounding, changed its cuticle conductance. This explains why the mango tissue infrared temperature was higher than 34.5 °C. The pedicel (Figure 8a) measured 32 °C, four degrees over the AC paste temperature as solar irradiation heat the pedicel.
The microphotography appears in line in materials and methods
Pedicel-peduncle samples were sampled on treated and control of approximately 2 cm each, which were fixed in 96 % ethanol:100 % glacial acetic acid (2:1; v:v) and processed in an automatic tissue exchanger (Tissuematon Fisher) with 2-ethoxyethanol (cellosolve) and xylene, then transfer to paraffin (55 °C) staying 72 hours inside a stove. The paraffin pyramid was made according to Sass (1968) technique and in a rotary microtome (model 820, American Optical, USA). Longitudinal cuts were made with a thickness of 10 µm. The cut sections were stained for 30 min at room temperature in a mixture of equal volumes of 0.1 % aqueous solutions of safranin and fast green, then washed in distilled water for 5 minutes and washed in 2 changes of absolute alcohol for 2–3 min [] (Bryan, 1955). The stained sections were then mounted on slides with coverslips by means of Haupt adhesive and 10 % formalin [] (Sass, 1968). Digital images were obtained and saved in jpg format with the aid of a Motic B3 Professional Series microscope, with the adaptation of a Moticam 480 camera with a 16 mm adapter.
The new microphotography image was added in Fig 10 showing each tissue
Previous to the figure an explanation is given:
In each of the four images, the left section is the one fixed to the fruit and the letter. Only a few letters were placed in figure 10, to show the different tissues and avoid data agglomeration. In the upper-right image, the microphotography shows the effect after applying a light AC paste, meanwhile in the lower-right a dense AC paste damages more the pedicel.
Figure 10. AC untreated pedicel-peduncle to the left and treated pedicel-peduncle on the right. C: Cortex that contains from outside to inside; epidermis (e), parenchyma (p), lactiferous channels (lc), sclerenchyma (s), parenchyma, phloem (ph) and cambium (ca). X: Xylem and pith. D: damage area that includes mainly the cortex and part of the xylem, where a large constriction is evident.
Line 315 Data source (Table2). Data were obtained from the second experiment with 90 mango fruits. The data results are shown in Table 2.
Line 366-369 concerning nets were moved to the beginning of the methodology section line 93. The two other paragraphs were let and a brief discussion of impacts caused by dropping the fruit are shown.
Fruits being harvested fall to the floor and affects its quality, as bruises result from the impact [38]. In experiments with green mature “Carabao” mango fruits in Philippines, 80% of the fruit that fell from a height of 5 m suffered cracks and 20% resulted bruised [39]. Fruit impacts results in cell wall and membrane rupture [40]. In our experiment the mesh avoided any crack or bruise.
Line 370-71 The discussion sentence was not found in ref 33. This reference was removed as the topic was removed from the discussion.
Section 4,2 does not look to these experiments. It was moved after the conclusion section and merely explains how in the future the UAV can detect the pedicel and apply the AC blend. It can be removed if the evaluators do not find it useful.
Line 419 UAV is Unmanned Aerial Vehicle

Reviewer 3 Report
Comments and Suggestions for Authors
The MS dealing with methods to improve the harvest efficiency of mango fruit in areas where mecanichal harvesters are unsuitables sounds interesting.
However some major flaws need to be solved before the paper could be considered for publication.
In particular, M&M section can be improved by adding some information and some sub-sections.
In Results section the figure 12 could be slightly modified and part of the Discussion need to be shortened and moved to Conclusion as perspective.
References numbering need also to be revised
Detailed comments and suggestion are in the attached pdf

Author Response
After the paper was revised the following points addressed:
The material and method section was revised and better ordered and how the microphotography’s taken. Information on the net was added.
Some references were removed and another added. The first one didn’t had the name and was addressed.
The discussion was changed moving the first paragraph to the beginning of material and methods. A brief discussion was added in the second paragraph with the thematic of impact which in this experiment didn’t occur due to the net. The third paragraph discuss the effect of the AC blend with fruit latex.
The future section was moved after the conclusion and considers how this technology can be applied in a farm in a future. It is not centered precisely on the topic but indicates which tools are required for its application.
All the corrections made by the reviewer in the text were considered.
PLEASE CHECK THE COMPLETE ARTICLE CHANGES AS SOME POINTS WERE ADDRESSED TO PREVIOUS REVIEWERS
Round 2
Reviewer 2 Report
Comments and Suggestions for Authors
In the first review, it was confirmed that most of the points were corrected.
However, in Fig. Please indicate the scale bar inside the picture in Figure 10. In microscope pictures, a scale bar must be displayed to estimate size.
Comments on the Quality of English LanguageI believe there is no problem with the English expression.
Author Response
The figure 10 has now the size limit within each picture.
As well all the red comments were taken away.
Please acknowledge the editors that in this page the name is lejandro which is OK in the article. The last author wants his naame published as Victor Prado-Hernandez

Reviewer 3 Report
Comments and Suggestions for Authors
Authors addressed the major questions and significantly revised the paper
Author Response
We fixed the figure and the red reviews are now normal in color.
In the top of this section the name of one researcher is not well written. It should be alejandro as in the article. The last author wants his naame to be published as Victor Prado-Hernandez